# Numerical Study on the Laser Annealing of Silicon Used in Advanced V-NAND Device

**DOI:** 10.3390/ma15124201

**Published:** 2022-06-13

**Authors:** Yeong-Il Son, Joonghan Shin

**Affiliations:** 1Department of Future Convergence Engineering, Kongju National University, Cheonan 31080, Korea; yil1554@naver.com; 2Department of Mechanical and Automotive Engineering, Kongju National University, Cheonan 31080, Korea

**Keywords:** laser annealing, numerical simulation, cell over periphery (COP), silicon, multipath, beam overlap

## Abstract

Laser melt annealing of amorphous silicon (a-Si) and subsequent recrystallization of a-Si are essential processes for successfully implementing vertical NAND (V-NAND) flash memory devices developed based on the cell-over-periphery (COP) structure. The aim of this study was to develop the numerical model for the laser melting process of a-Si used in V-NAND COP structure. In this study, the numerical simulation predicting the temperature distribution induced by multipath laser scanning and beam overlapping was conducted. In particular, the temperature uniformity and melt duration issues, which are critical in practical laser melt annealing applications in semiconductor fabrication, were discussed based on the simulated temperature distribution results. According to the simulation results, it was found that the annealed surface was subjected to rapid heating and cooling. The heating and cooling rates after temperature stabilization were 4.7 × 10^7^ K/s and 2.04 × 10^7^ K/s, respectively. The surface temperature increased with time and beam overlap ratio owing to the preheating effect and increasing heat accumulation per unit area. Under the process conditions used in the simulation, the temperature in a-Si was far above its melting point (1440 K), which numerically indicated full melting of the a-Si layer. Temperature uniformity within the annealed area was significantly improved when an overlap ratio of 50% was used. It was also found that using an overlap ratio of 50% increased the melt duration by 29.8% compared with an overlap ratio of 25%.

## 1. Introduction

With advances in semiconductor manufacturing technology, the bit density of NAND flash memory has increased over the last few decades. Moreover, the technology node for next-generation devices has now reached several nanometers. In the past, NAND flash memories were manufactured by integrating transistors in a two-dimensional plane; however, this method has a limitation regarding production of high-performance semiconductors. To overcome this limitation, vertical NAND (V-NAND) flash memory was developed. V-NAND is a three-dimensional structure in which more than 100 two-dimensional plane NANDs are stacked; it can store much more data than two-dimensional NAND [1,2,3,4]. Recently, a novel V-NAND structure using the cell-over-periphery (COP) scheme has been proposed. In general, because the cell and periphery areas coexist on the same plane, the cell density of V-NAND devices can only be increased in a limited manner. However, a V-NAND using the COP scheme forms a circuit by separating the cell and the peripheral areas into different stacks. As a result, more cells can be integrated onto a wafer, which leads to better device performance [5,6]. Using the COP scheme, high-quality polycrystalline silicon (poly-Si), substituting the single-crystal silicon (sc-Si) substrate in the conventional V-NAND structure on the insulator (a thick oxide layer), is essential to successfully form a V-NAND cell circuit. The high-quality poly-Si layer is obtained through the recrystallization of amorphous silicon (a-Si) or deposition of low-quality poly-Si layer onto an insulating oxide layer. Therefore, the proper annealing process to induce recrystallization of the deposited silicon layer is crucial in the V-NAND COP scheme. Rapid thermal processing (RTP) [7,8], a conventional annealing method, has been widely used in the fabrication of semiconductor devices. However, the heat budget of the RTP is fairly large for a scaled device with an ultra-shallow junction (USJ), as the annealing time of the RTP is usually in the order of tens of seconds. To reduce the heat budget, flash lamp annealing (FLA) and laser annealing (LA) have been developed. The annealing time (millisecond level) in FLA is much shorter than that of RTP, resulting in a significantly reduced heat budget. However, because the heat sources used to heat the front and back sides of the wafer are different (a Xe flash lamp is used for front-side heating, while a halogen lamp is used for back-side heating), an extreme temperature gradient occurs between the front and back side of the wafer. Usually, the temperature of the front side of the wafer is approximately 300–400 K higher than that of the back side. This large temperature gradient causes various problems, such as crystalline defects, warpage, and fracturing of the wafer [9,10]. Regarding the LA process, an annealing time of less than a millisecond is possible, which in turn can minimize the heat budget. Because the RTP and FLA processes require a relatively low radiation energy level intensity, they are only suitable for non-melting processes such as dopant activation. On the other hand, the LA process, which uses a laser beam of high intensity, is not only suitable for non-melting processes but also for melting processes such as laser-induced recrystallization [11,12,13,14,15,16,17]. In addition, wafer warpage is relatively rare in the LA process because the wafer is locally heated by a small laser beam spot.

Owing to these advantages, the LA process has recently attracted increasing attention in the semiconductor fabrication field. Park et al. [18] conducted a comprehensive study of blue diode laser annealing of a-Si thin films. It was found that annealing using a blue diode laser facilitated the melting of a-Si owing to the greater level of heat diffusion and deeper penetration depth. Sakaike et al. [19] studied the effect of a diode laser on the crystallization of amorphous germanium (a-Ge) films. The scan speed required to melt and recrystallize a-Ge films was confirmed through an in-situ monitoring of the process. Kim et al. [20] reported a poly-Si thin film transistor (TFT) fabricated via excimer laser recrystallization. They produced large poly-Si grains exceeding 4 μm by reducing the solidification velocity of the melted Si owing to the adoption of an air-gap structure with low thermal conductivity. Chowdhury et al. [21] used excimer laser sources with various wavelengths (248, 266 and 308 nm) for the crystallization of hydrogenated a-Si. In their study, all excimer lasers successfully melted a-Si layers of different thicknesses (20–80 nm) without causing thermal damage to the substrate. Moreover, they found that the crystallinity obtained by using an excimer laser of 266 nm was similar to that produced with conventional thermal annealing. In addition to the studies mentioned above, various modeling studies regarding the LA process have been carried out to gain a deeper understanding of the fundamental physics behind the process. Shih et al. [22] numerically simulated the excimer laser crystallization processing of thin Si films on oxide substrates. Through simulation, they deduced numerous important outcomes (temperature, melt depth, and duration and position of the solidification interface) describing the crystallization process. Thoedorakos et al. [23] modeled the picosecond laser-annealing process of an a-Si thin film used in a solar cell. The crystallinity ratio and crystal size of the poly-Si thin film, which improved the performance of the solar cell, were investigated numerically. Garcia et al. [24] carried out a numerical simulation of the single-pulse laser annealing of a-Si thin films. In their study, the annealed surface temperature was numerically determined to support the experimental results of the crystalline fraction. For 355 and 532 nm lasers, the crystalline fraction obtained experimentally agreed reasonably well with the simulation results. Degorce et al. [25] reported a three-dimensional transient temperature field model for laser annealing. The temperature distribution created by a single laser pulse was determined for the SiO_2_/Si bilayers. In addition, information such as the velocity of the melt interface and diameter of the melt surface was numerically obtained.

To date, many researchers have conducted numerical studies on LA processes with different goals. However, the outcomes of most simulation studies are somewhat limited by the restrictive simulation assumptions such as one laser shot or a fixed beam position. These conditions can differ considerably from the actual process conditions. In an actual LA process, multi-scanning and beam overlapping are essential to achieve uniform heating at the wafer-level. In this study, the LA process for a-Si used in the COP structure was numerically simulated. In particular, a numerical simulation considering the heat transfer phenomena induced by multipath scanning and beam overlapping in a three-dimensional multilayer structure was performed. The temporal and spatial temperature distributions during the multipath laser scanning process were studied. The effects of beam overlapping on the temperature uniformity were also investigated. Finally, the melt duration during the LA process was discussed.

## 2. Development of a Numerical Model

### 2.1. Model Geometry and Process Conditions

For the numerical simulation, a commercial software (COMSOL Multiphysics) based on the finite element method was used. Figure 1 shows the geometrical information of the computational domain used in the simulation. As shown in the figure, the computational domain was developed to replicate the COP structure, and a fine mesh was adopted for the area scanned by a laser beam.

This study mainly aimed to investigate the temperature distribution during multipath scanning of a laser beam. To achieve this, the scan path of the laser beam was set to follow a zigzag pattern, as shown in Figure 2. The intensity of a laser beam in this simulation has a Gaussian distribution (see Equation (3)). Therefore, in order to reduce the non-uniformity of heating during the multipath scanning of a laser beam, the overlap of a laser beam is essential. In this simulation, three overlap conditions (0%, 25% and 50% overlap ratio) not exceeding the radius of a laser beam spot were selected, as shown in Figure 2. An overlap ratio higher than 50% was not considered because it could induce an overheating zone between two adjacent scanning paths. The process conditions adopted for the simulation are presented in Table 1. It was assumed that the type of laser is a continuous-wave mode.

### 2.2. Governing Equation and Boundary and Initial Conditions

In this simulation, the phase change induced by the melting process was considered; however, the liquid flow that may have been produced by melting was not. The governing equation to obtain the temporal and spatial temperature distributions within the computational domain is expressed as
(1)$$\rho C_{p}\frac{\partial T}{\partial t}-\nabla \left(k\nabla T\right)=Q$$
where ρ is the density, $C_{p}$ is the specific heat, and k is the thermal conductivity of the material. The right-hand side of Equation (1) represents the volumetric heat source related to the absorption of the laser beam in the material. The detailed expression of the volumetric heat source Q can be expressed using the Beer–Lambert law [26] as follows:(2)$$Q=I\cdot \left(1-R\right)\cdot \alpha \cdot \exp(-\alpha \left|z\right|)$$

Here, R is the reflectivity and α is the absorption coefficient. I indicates the beam intensity with a Gaussian distribution, which is determined using the following equation
(3)$$I=I_{0}\cdot \exp\left(\frac{-\left({\left(x-X_{n}\right)}^{2}+{\left(y-Y_{n}\right)}^{2}\right)}{2r^{2}}\right)$$
where $X_{n}$ and $Y_{n}$ are the *x* and *y* directional coordinates of the beam center that change over time, respectively, and $I_{0}$ is the peak intensity of the laser beam, which can be expressed as follows:(4)$$I_{0}=\frac{2P}{\pi r^{2}}$$

Here, P is the laser power and r is the beam radius.

The phase change effect for melting is considered in the simulation by employing the enthalpy method [27], as follows:(5)$$C_{p}\left(T\right)={\tilde{C}}_{p}\left(T\right)+g\left(T\right)L_{m}$$
where ${\tilde{C}}_{p}\left(T\right)$ is the specific heat as a function of temperature, $L_{m}$ is the latent heat of fusion, and $g\left(T\right)$ is the Gaussian distribution function, which is defined as follows:(6)$$g\left(T\right)=\frac{1}{\Delta T\sqrt{\pi }}\exp\left(-\left(\frac{{\left(T-T_{m}\right)}^{2}}{\Delta T^{2}}\right)\right)$$

In this equation, ΔT (30 K) denotes the temperature range corresponding to the phase transition and, $T_{m}$ denotes the melting point of a-Si.

To solve Equation (1), suitable boundary conditions should be defined. For the top surface, convection and radiation heat losses are considered as boundary conditions as follows:(7)$$-n\cdot \left(k\nabla T\right)=h\left(T-T_{\infty }\right)+\varepsilon \sigma \left(T^{4}-T_{\infty }{}^{4}\right)$$
where n is the surface normal vector, h is the convective heat transfer coefficient, ε is the surface emissivity, σ is the Stefan–Boltzmann constant, and $T_{\infty }$ is the ambient temperature (298 K).

Because the size of the computational domain is sufficiently large compared with that of the laser beam, we can assume that no significant temperature change occurs across the edge areas of the computational domain. Thus, the following adiabatic boundary condition can be applied to all side areas (vertical faces) of the computation model.
(8)$$-n\cdot \left(k\nabla T\right)=0$$

During the LA process, a wafer is usually heated using a heating chuck placed under the wafer, which can enhance the absorption rate of the laser beam in the wafer. In this study, we assumed that the entire wafer is preheated to 673 K using the heating chuck. Therefore, the initial temperature of the wafer was set to 673 K in the computational model. Owing to using the heating chuck, this fixed temperature (673 K) assumption was also imposed on the bottom surface of the computational domain. The constant values used in the boundary conditions are listed in Table 2.

### 2.3. Material Properties

The thermophysical and optical properties of a-Si were expressed as a function of temperature to reflect more realistic physical phenomena. In this simulation, the laser beam wavelength was set to 532 nm, which is often used in the silicon recrystallization process [28,29]. Therefore, optical properties corresponding to 532 nm were selected for the simulation.

The thermophysical and optical properties of the top a-Si layer used in the simulation are presented in Table 3. For the thermophysical properties of silicon dioxide (SiO_2_) and substrate silicon (sub-Si), constant values reported in the COMSOL Multiphysics database were used. Based on Equation (2) and optical properties presented in Table 3, it was found that the energy of the laser beam was absorbed almost entirely (~99%) after reflection on a-Si surface within the a-Si layer. Therefore, the effect of beam reflection at the subsequent interfaces of a-Si/SiO_2_ and SiO_2_/sub-Si was not considered in this study. 

## 3. Results and Discussion

### 3.1. Temporal Profile of the Surface Temperature

Figure 3 illustrates the temporal profile of the temperature obtained at five points on the first scan path. In the simulation, the center of the laser beam moved along the scan path. As shown in the figure, the temperatures at all points underwent rapid heating and cooling as the laser beam passed through. The heating rate was the most significant around point 1, which was the starting point for annealing. The heating rate at point 1 was 8.95 × 10^7^ K/s, which was 88.8% higher than that at point 2 (4.7 × 10^7^ K/s). The maximum temperature was found to be sufficiently high to melt the top a-Si layer. It was also found that the maximum temperature and temperature profile did not change much between points 2 and 5. This led to a similar heating rate between points 2 and 5. The increase in temperature between points 1 and 2 was due to the preheating of the laser beam after it passed point 1. As shown in Figure 4, by the time the laser beam passes around the position (at 80 μs) near point 2, point 2 was already preheated to 1205 K (see the dotted circle in the figure). This preheating quickly stabilized and continued until the end of the scan path, contributing to similar maximum temperatures at points 3, 4, and 5. The cooling rate was typically lower than the heating rate, as shown in the figure. This feature of the temperature profile generated by LA was also presented in [34]. The cooling rates for points 1 and 2 were 2.82 × 10^7^ K/s and 2.04 × 10^7^ K/s, respectively. The cooling rate calculated in this study is considerably lower than that (8–9 × 10^9^ K/s) of pulsed LA presented in [35]. The annealing time (50 μs, beam diameter divided by scan speed) used in the simulation is longer than that (23 ns) of the above pulse LA process. This long annealing time induced the relatively low cooling rate. It was also observed that the temperature decrease at point 5 was somewhat gradual. This is attributable to the greater accumulation of the laser energy around the corner area.

Figure 5 shows the temperature change over time under different beam overlap conditions during multipath annealing. The temperature profiles shown in the figure were obtained from the center of the moving laser beam. As shown in the figure, the temperature gradually increased with time. Around the starting point of the first scan path, the temperature increased significantly; however, it stabilized rapidly as the preheating effect decreased. At the end of the first scan path, another considerable temperature increase was induced as the laser beam moved along the y-axis. This was attributable to the heat accumulation at the corner area of the scan path. This increase in temperature generated hot spots around the corner areas of the scan path, as indicated in the figure. These hot spots produced during the annealing process worsen the temperature non-uniformity phenomenon, which in turn can lead to various defects in the semiconductor device. Therefore, a process condition that prevents the non-uniform temperature distribution should be chosen. As shown in the figure, the higher the overlap ratio was, the steeper the temperature gradient near the hot spot became. The increase in the overlap ratio confined more heat to the corner area given that the distance between two adjacent scan paths was reduced.

### 3.2. Temperature Distribution in the Depth Direction

The temperature distribution along the depth direction in a multilayer COP structure was investigated as well. Figure 6 shows a schematic of the cross-section of the computational domain and temperature distribution in the depth direction of the computational domain. The temperature distribution is obtained at the midpoint of the scan path in each case as depicted in the figure (the second scan path for 0% overlap ratio and the third scan path for 25% and 50% overlap ratios). As shown in the figure, the temperature decreased almost linearly with an increasing depth. However, the temperature gradient in a-Si was relatively larger than that in the SiO_2_ layer because the laser energy was mostly absorbed in the top a-Si layer. According to the simulation results, the temperatures in the a-Si layer remained far above its melting point (1440 K) at all overlap ratios. This implies that the a-Si can be completely melted and recrystallized under the process conditions used in the simulation. According to the study of Xu et al. [36], the beam intensity of 3.9 × 10^7^ W/cm^2^ was required to melt the Si of 350 nm in the nanosecond pulsed LA process. However, in this study, relatively low beam intensity (1.5 × 10^5^ W/cm^2^) was enough to melt the Si of 400 nm. This was attributed to the long annealing time (25 μs) of CW LA compared with the nanosecond-pulsed LA (annealing time of 26 ns).

The temperatures that were obtained at the interface between a-Si and SiO_2_ were 1822 K, 1944 K, and 1958 K for overlap ratios of 0%, 25%, and 50%, respectively. These temperatures were lower than the melting point (1983 K) of SiO_2_. Notice that under this condition, the SiO_2_ layer remained sufficiently solid to support the melted Si at the top.

### 3.3. Analysis of the Temperature Uniformity with Different Overlap Ratios

In this section, the temperature uniformity, as indicated by the temperature difference within the scanned area, induced with different overlap ratios is discussed. Figure 7 shows the schematics of the beam-scanning path and surface temperature evolution. The temporal profiles of the temperature were obtained at the points marked along the beam scanning direction in the figure. Throughout the beam-scanning path, the dotted lines indicate the overlapping areas between two adjacent scanning paths. The temperature profiles for the overlapped areas and scanning paths are indicated by the dotted and solid lines in Figure 7, respectively.

As mentioned earlier, a laser beam with a Gaussian intensity distribution was considered in this simulation. The characteristics of the Gaussian beam distribution inevitably generate a temperature difference within the beam spot. This leads to a non-uniform temperature distribution in the scanning area during the annealing process. Because a severely non-uniform temperature distribution can cause defects in a semiconductor device, it should always be avoided. In an LA process, this non-uniform temperature distribution can be alleviated by beam overlapping. Figure 7 clearly indicates that the temperature difference between the overlapped area and scanning path decreases with an increase in the overlap ratio. At an overlap ratio of 0%, the maximum temperature difference at point 3 was very large (approximately 1200 K); hence, the temperature of the overlapped area could not reach the melting point (1440 K). However, at an overlap ratio of 50%, the temperature difference between the two regions drastically decreased. The temperature difference was approximately 282 K, which indicates that the temperature uniformity was improved owing to beam overlapping. As presented in the figure, the temperature at the beam overlapping area is usually lower than that of the scanning path because it is close to the beam edge. Based on the simulation results, at overlap ratios of 25% and 50%, it is confirmed that the maximum temperature of the overlapped area was higher than the melting point (1440 K) of a-Si. This implies that in all areas, including the beam-overlapping zone, the a-Si melted sufficiently during the annealing process. 

### 3.4. Analysis of Melt Duration

The electrical performance of a semiconductor device can be enhanced by increasing the polycrystalline material grain size [37]. For the COP structure, increasing the grain size of the recrystallized Si layer is also important for obtaining a high performance COP device. In the liquid-phase recrystallization process, a long melt duration induced by high temperature and slow cooling rate is generally advantageous for producing large polycrystalline grains by suppressing the random formation of nucleation sites during grain growth [38].

Figure 8 demonstrates the melt duration with the overlap ratio, determined from the results shown in Figure 7. Figure 8 indicates that the average melt duration increased with an increase in the overlap ratio. At an overlap ratio of 0%, the melt duration of the overlapped area could not be measured because the temperature of that area did not reach the melting point. At an overlap ratio of 50%, the melt duration increased by 29.8% when compared with the 25% overlap condition. Based on this result, we predict that under the 50% overlap condition, a considerable increase in the grain size can be achieved.

## 4. Conclusions

In this study, the laser melting process for a-Si used in the V-NAND COP structure was numerically simulated. The major conclusions of this study are as follows.

(1)The annealed surface was subjected to rapid heating and cooling. The heating and cooling rates after temperature stabilization were 4.7 × 10^7^ K/s and 2.04 ×10^7^ K/s, respectively.(2)The surface temperature gradually increased with time and beam overlap ratio because of the preheating effect and greater heat accumulation per unit area. More heat accumulation per unit area also induced hot spots at every corner of the scanning path.(3)Under the process conditions used in the simulation, the entire a-Si area was completely melted. The temperatures (1822–1958 K) at the interface between a-Si and SiO_2_ were lower than the melting point (1983 K) of SiO_2_. Thus, the SiO_2_ layer remained sufficiently solid to support the melted Si at the top(4)The temperature uniformity in the annealed area was considerably improved via beam overlapping. It was also found that the melt duration of the annealed a-Si increased with an increase in the overlap ratio. This indicated that a considerable increase in grain size could be achieved using beam overlapping.(5)In present study, the experimental work to validate the numerical model was not conducted, and this is the weakness of this study. As a future study, we are planning to develop a relevant validation method for our numerical model (i.e., the measurement of surface reflectivity or grain size for molten Si layer).

## Figures and Tables

**Figure 1 materials-15-04201-f001:**
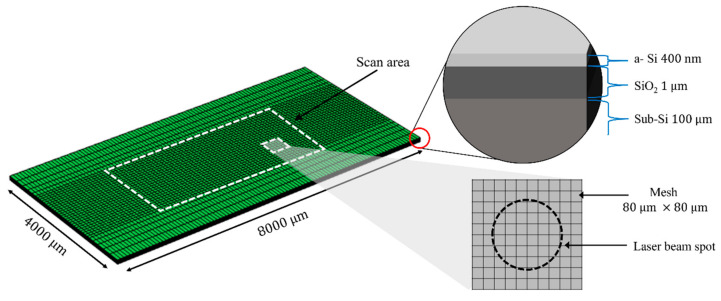
Computational domain used in simulation.

**Figure 2 materials-15-04201-f002:**
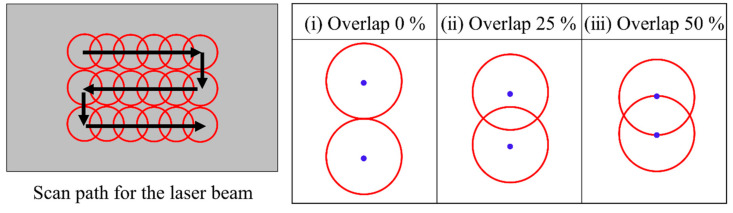
Scan path and overlap ratio for the laser beam, where black arrows indicate the scanning direction of the laser beam. The red circles and blue points in the red circles denote the edge and center of the laser beam, respectively.

**Figure 3 materials-15-04201-f003:**
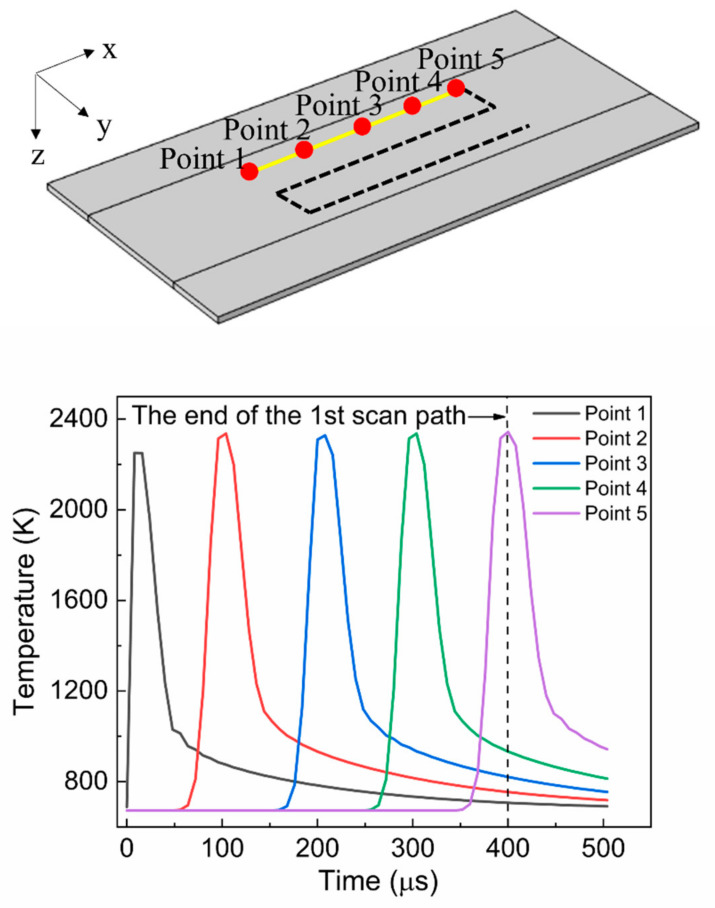
Temporal profile of the temperature obtained at five points over the first scan path.

**Figure 4 materials-15-04201-f004:**
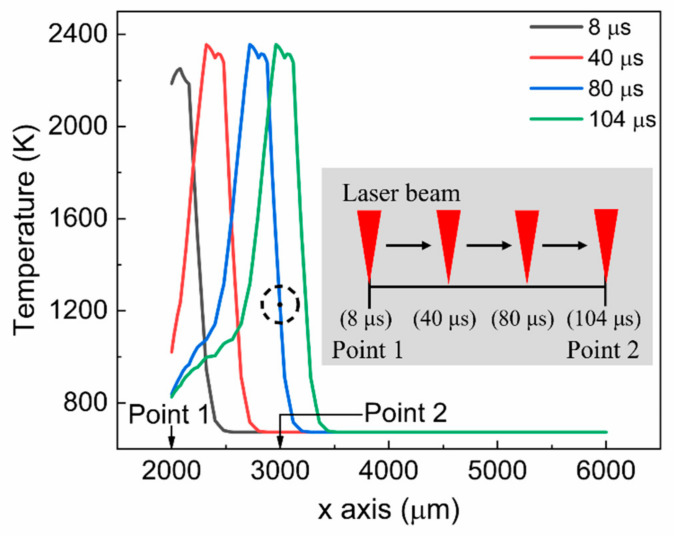
Spatial temperature distribution at different times.

**Figure 5 materials-15-04201-f005:**
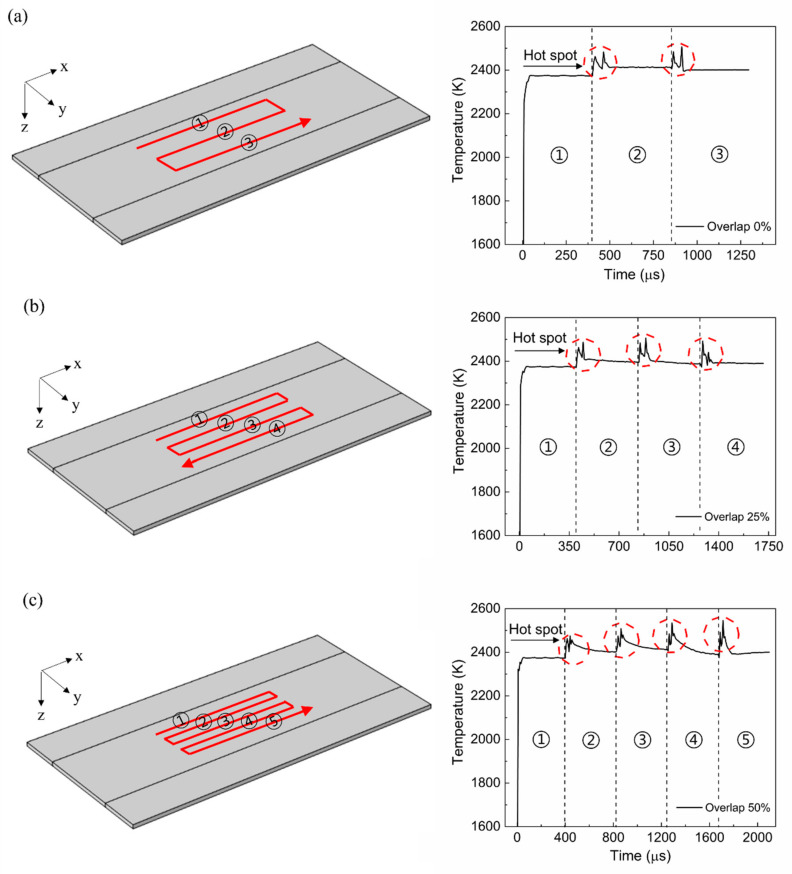
Schematic of the laser scan path and temporal profile of the surface temperature: (**a**) 0% overlap, (**b**) 25% overlap, and (**c**) 50% overlap.

**Figure 6 materials-15-04201-f006:**
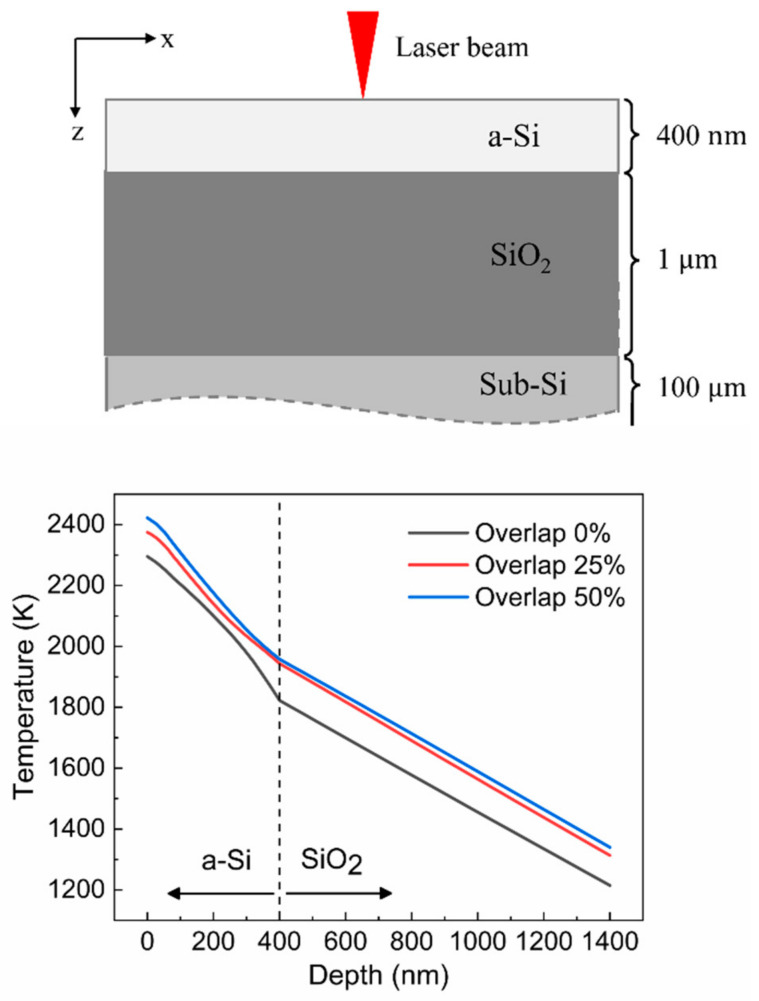
Cross-section of the computational domain and temperature distribution in the depth direction.

**Figure 7 materials-15-04201-f007:**
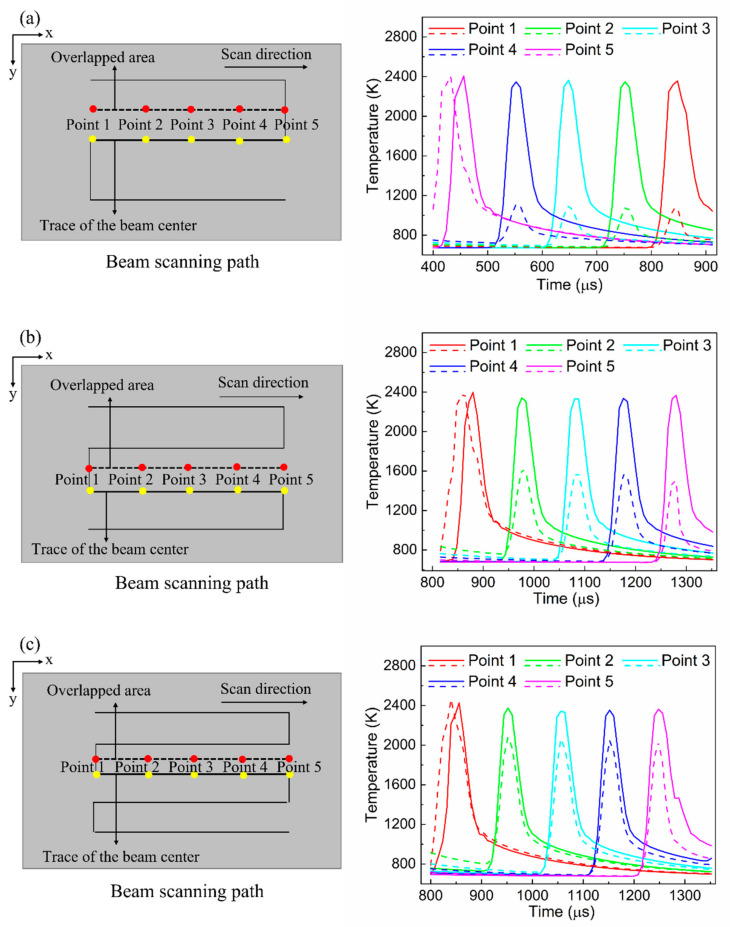
Schematics of beam-scanning path (each pair of red and yellow spot at a particular point has the same coordinate on the y-axis) and surface temperature evolution: (**a**) 0% overlap ratio, (**b**) 25% overlap ratio, and (**c**) 50% overlap ratio. Dotted lines and solid lines indicate the temperature profiles at red spots (overlapped areas) and yellow spots (scanning paths), respectively.

**Figure 8 materials-15-04201-f008:**
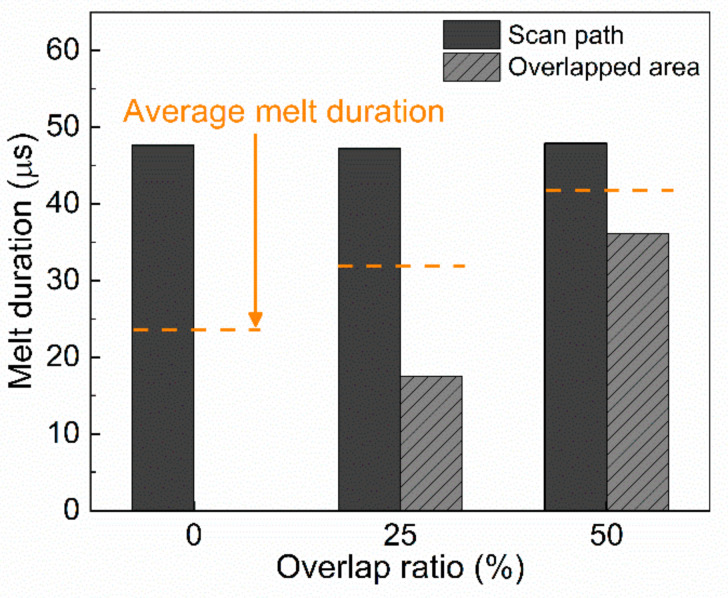
Melt duration (at point 3 in Figure 7) for different overlap ratio.

**Table 1 materials-15-04201-t001:** Annealing process conditions used in the simulation.

Process Condition	Symbol	Unit	Value
Beam radius	r	μm	250
Scan speed	V	$m\cdot s^{-1}$	10
Laser power	P	W	300
Average beam intensity	I	$W\cdot {\mathrm{cm}}^{-2}$	$1.5\times 10^{5}$

**Table 2 materials-15-04201-t002:** Constant values used in boundary condition.

Constant	Unit	Value	Description
h	$W\cdot m^{-2}\cdot K^{-1}$	2	Convective heat transfer coefficient
ε	–	0.4	Surface emissivity
σ	$W\cdot m^{-2}\cdot K^{-4}$	$5.67\times 10^{-8}$	Stefan-Boltzmann constant

**Table 3 materials-15-04201-t003:** Thermophysical and optical properties used in the simulation.

Property	Symbol	Unit	a-Si	SiO_2_	Sub-Si
Melting temperature	$T_{m}$	K	1440	–	–
Latent heat of fusion	$L_{m}$	$J\cdot {\mathrm{kg}}^{-1}$	$1.320\times 10^{6}$ [22]	–	–
Thermal conductivity	k	$W\cdot m^{-1}\cdot K^{-1}$	1.3$\times 10^{-9}{\left(T-900\right)}^{3}$+ 1.3$\times 10^{-7}{\left(T-900\right)}^{2}$+ 1$\times 10^{-4}\left(T-900\right)$+ 1 for (298 < T< $T_{m}$), 0.7 for ($T_{m}$< T< 2500) [30]	1.4	34
Specific heat	$C_{p}$	$J\cdot {\mathrm{kg}}^{-1}\cdot K^{-1}$	852.28 + 0.08791 T− 0.0014751(1/$T^{2}$) for (298 < T< $T_{m}$), 852.28 + 0.08791 T− 0.0014751(1/$T^{2}$) + g(T)$L_{m}$ for ($T_{m}-\cdot T/2$< T< $T_{m}$+ ·T/2), 968.22 for ($T_{m}$+ ·T/2< T< 2500) [31,32]	730	678
Density	ρ	$\mathrm{kg}\cdot m^{-3}$	2440–0.0544T for (298 < T< 1000), 2524 for (1000 < T< $T_{m}$) 2524–0.3487 (T− 1683) for ($T_{m}$< T) [32]	2200	2320
Absorption coefficient (at 532 nm)	α	${\mathrm{cm}}^{-1}$	$1.25\times 10^{5}$ for (298 < T< $T_{m}$), $1\times 10^{6}$ for ($T_{m}$< T< 2500) [29,33]	–	–
Reflectivity (at 532 nm)	R	–	0.4 for (298 < T< $T_{m}$), 0.73 for ($T_{m}$< T< 2500) [33]	–	–

## Data Availability

Data are available from the corresponding authors upon reasonable request.

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
