# Peer review of "Numerical Study on the Laser Annealing of Silicon Used in Advanced V-NAND Device"

_materials, 2022, doi:10.3390/ma15124201_

Round 1

Reviewer 1 Report

The authors used numerical simulation to study the laser melt process of the a-Si, and the evolution of the temperature distribution by laser scanning was predicted. According to the simulation results, the authors concluded that the overlap ratio of beam spots was critical for the temperature increasing and uniformity of melting. The results and conclusions are beneficial for practical application of laser processing in semiconductor fabrication. The authors simulated the temperature distribution in laser scaning. The results and conclusions are important for the laser ablation in real applications and may be helpful for understanding the phase changing of Si induced by laser. In addition, the authors provided enough data and evidences to support their conclusions.The manuscript was well written and I recommend to accept the manuscript after a minor revision:

  1. In Figure 3 (right figure), I think the “time” should be “x axis” or the unit should be “μs” for x-axis? Please clarify.
  2. In simulation, the laser source was pulsed or continuous wave?
  3. In line 187, it seems Ref. [25] is cited by mistake, because in Ref. [25], 532 nm laser was not mentioned. Please confirm.

Author Response

To editors and reviewers:

We really appreciate your valuable comments. We have finished the revision and also made responses to the comments of reviewers. The detail contents of the revision and response are described below (the edited parts are colored ‘red’ in the revised manuscript).  

COMMENTS FOR THE AUTHOR:

Reviewer 1

Comments and Suggestions for Authors

The authors used numerical simulation to study the laser melt process of the a-Si, and the evolution of the temperature distribution by laser scanning was predicted. According to the simulation results, the authors concluded that the overlap ratio of beam spots was critical for the temperature increasing and uniformity of melting. The results and conclusions are beneficial for practical application of laser processing in semiconductor fabrication. The authors simulated the temperature distribution in laser scaning. The results and conclusions are important for the laser ablation in real applications and may be helpful for understanding the phase changing of Si induced by laser. In addition, the authors provided enough data and evidences to support their conclusions.The manuscript was well written and I recommend to accept the manuscript after a minor revision:

  1. In Figure 3 (right figure), I think the “time” should be “x axis” or the unit should be “μs” for x-axis? Please clarify.

Ans) The unit for time in Fig. 3 has been changed to “μs”

  1. In simulation, the laser source was pulsed or continuous wave?

Ans) The laser source was continuous wave (CW). The information for laser type was added to the revised manuscript (line 142 in page 5).

  1. In line 187, it seems Ref. [25] is cited by mistake, because in Ref. [25], 532 nm laser was not mentioned. Please confirm.

Ans) The Ref. [25] mentioned above has been substituted by the right reference (Ref. [25] and [26] in the revised manuscript) (line 197 in page 7). Because of the addition of new reference and the rearrangement of references, new numbering for the reference has been performed.

Reviewer 2 Report

The work is very weak, it does not bring anything new in the field. The authors did not present any novelty arguments. The part of the results and discussions is very poorly treated, few analyzes and results uncorrelated with the literature. To fully recover and resubmit.

Author Response

To editors and reviewers:

We really appreciate your valuable comments. We have finished the revision and also made responses to the comments of reviewers. The detail contents of the revision and response are described below (the edited parts are colored ‘red’ in the revised manuscript).  

Reviewer 2

Comments and Suggestions for Authors

The work is very weak, it does not bring anything new in the field. The authors did not present any novelty arguments. The part of the results and discussions is very poorly treated, few analyzes and results uncorrelated with the literature. To fully recover and resubmit.

Ans) As described in the last paragraph of the introduction, the novelty of this paper is in the development of the numerical model that can handle the heat transfer phenomena induced by multipath scanning and beam overlapping in a three-dimensional multilayer structure during the laser annealing process. It is expected that this numerical model could reasonably predict the actual laser annealing process in a wafer level (so far, the outcomes of most simulation studies for laser annealing are somewhat limited in that the simulations employed restrictive assumptions such as one laser shot, a fixed beam position or two-dimensional model). Extensive English editing for the whole text has been conducted.

Reviewer 3 Report

The paper reports a study on numerical simulation of the laser melt process of the a-Si and prediction of the temperature distribution in multipath laser scanning and beam overlapping. The data are valuable and interesting to readers in the field.  The paper is well written and can be accepted for publication. However, the author shall check and clarify the following points: 

  1. In Fig. 3, the unit for Time is not correct.
  2. What is the consideration when choosing the pre-heating temperature of 673K ? 

Author Response

To editors and reviewers:

We really appreciate your valuable comments. We have finished the revision and also made responses to the comments of reviewers. The detail contents of the revision and response are described below (the edited parts are colored ‘red’ in the revised manuscript).  

Reviewer 3

Comments and Suggestions for Authors

The paper reports a study on numerical simulation of the laser melt process of the a-Si and prediction of the temperature distribution in multipath laser scanning and beam overlapping. The data are valuable and interesting to readers in the field.  The paper is well written and can be accepted for publication. However, the author shall check and clarify the following points: 

-. In Fig. 3, the unit for Time is not correct.

Ans) The unit for time has been changed to “μs”

-. What is the consideration when choosing the pre-heating temperature of 673K ? 

Ans) As described in section 2.2., wafer preheating temperature of 673 K was assumed in the simulation. Absorption rate of a laser beam in material (Si) can be considerably increased by increasing the temperature of the material (efficient use of the laser energy is possible). Because of this, in the real semiconductor industry, the wafer is usually preheated with an AlN heater (or called AlN hot chuck) placed below the wafer during the laser annealing process. The preheating temperature of 673 K has been widely used in the industry because this temperature is considered the maximum temperature that can be applied to the laser annealing process while guaranteeing the stable use of the AlN heater. In this study, we imitated this wafer preheating condition.

Reviewer 4 Report

A very interesting article with presented practical applications. The article is clear, the methods used are comparable, the obtained results are correctly interpreted. The weakness of the article is the lack of validation of the numerical model.

Author Response

To editors and reviewers:

We really appreciate your valuable comments. We have finished the revision and also made responses to the comments of reviewers. The detail contents of the revision and response are described below (the edited parts are colored ‘red’ in the revised manuscript).  

Reviewer 4

Comments and Suggestions for Authors

A very interesting article with presented practical applications. The article is clear, the methods used are comparable, the obtained results are correctly interpreted. The weakness of the article is the lack of validation of the numerical model.

Ans) We have a plan to conduct experimental study for laser annealing and validation work for the numerical model in the near future. Generally, the direct measurement of the temperature distribution during the laser annealing process is very difficult due to the small beam spot (below 1 mm in size) and high scan speed of the laser beam (~10 m/s). Instead of the temperature measurement, we are considering indirect validation methods such as the melt duration measurement using surface reflectivity or the grain size analysis of recrystallized Si layer using EBSD (electron backscatter diffraction). The discussion for the future work plan is newly added to conclusion section (line 326-332 in page 15).

Reviewer 5 Report

The work investigated the numerical investigations on temperature distribution of laser annealed silicon which is used for semiconductor industries. The presented work is good and interest for readers. However, following shortcomings are presented in the manuscript and which has to be rectified before further processing. I request mandatory revision, as listed below, please do not simply respond but revise manuscript.

·         The equations should be formulated using Mathtype® software instead of Microsoft equation tools.

·         On what basis the ranges of overlap % has been selected. A detailed explanation and justification should be included.

·         Authors should explain about the feasibility of converting numerical investigation into experimental work for an accurate prediction of laser beam influence on temperature distribution. Because, the numerical investigations provided only an approximate result, whereas experimental study will provide accurate results.

·         For readers to quickly catch the contribution in this work, it would be better to highlight major difficulties and challenges, and authors' original achievements to overcome them, in a clearer way in Introduction section.

·         It is suggested to highlight the limitations of this study, suggested improvements of this work and future directions in the conclusion section. Also, the conclusion can be presented better than the present form with more findings.

Please note that the comments are intended merely to assist the authors in improving the manuscript and ensuring that published papers are of the highest quality. They are in NO WAY intended to discourage or demean the authors personally.

Author Response

To editors and reviewers:

We really appreciate your valuable comments. We have finished the revision and also made responses to the comments of reviewers. The detail contents of the revision and response are described below (the edited parts are colored ‘red’ in the revised manuscript).  

Reviewer 5

Comments and Suggestions for Authors

The work investigated the numerical investigations on temperature distribution of laser annealed silicon which is used for semiconductor industries. The presented work is good and interest for readers. However, following shortcomings are presented in the manuscript and which has to be rectified before further processing. I request mandatory revision, as listed below, please do not simply respond but revise manuscript.

  • The equations should be formulated using Mathtype® software instead of Microsoft equation tools.

Ans) The all equations in the manuscript were formulated using Mathtype® software

  • On what basis the ranges of overlap % has been selected. A detailed explanation and justification should be included.

Ans) The intensity (or power) of the laser beam in our simulation has a Gaussian distribution. Therefore, in order to reduce the non-uniformity of heating during the multipath scanning of a laser beam, the overlap of the laser beam is essential. In the simulation, two overlap conditions (25 and 50% overlap ratio) not exceeding the radius of a laser beam spot were selected (overlap ratio higher than 50 % may induce an overheating zone between two adjacent scanning paths), and the results of the simulation using these conditions were compared with those of the simulation using non-overlapped condition. The additional explanation for the overlap condition of a laser beam has been included in the revised manuscript (line 136-141 in page 5).    

  • Authors should explain about the feasibility of converting numerical investigation into experimental work for an accurate prediction of laser beam influence on temperature distribution. Because, the numerical investigations provided only an approximate result, whereas experimental study will provide accurate results.

Ans) In the present study, numerical simulation to obtain temperature distribution was only conducted. However, we have a plan to conduct experimental study for laser annealing and validation work for the numerical model in the near future. Generally, the direct measurement of the temperature distribution during the laser annealing process is very difficult due to the small beam spot (below 1 mm in size) and high scan speed of the laser beam (~10 m/s). Instead of the temperature measurement, we are considering indirect validation methods such as the melt duration measurement using surface reflectivity or the grain size analysis of recrystallized Si layer using EBSD (electron backscatter diffraction). The discussion for the future work plan is newly added to conclusion section (line 326-332 in page 15).

  • For readers to quickly catch the contribution in this work, it would be better to highlight major difficulties and challenges, and authors' original achievements to overcome them, in a clearer way in Introduction section.

Ans) As described in the last paragraph of the introduction, the novelty (major contribution) of this paper is in the development of the numerical model that can handle the heat transfer phenomena induced by multipath scanning and beam overlapping in a three-dimensional multilayer structure during the laser annealing process. It is expected that this numerical model could reasonably predict the actual laser annealing process in a wafer level. So far, the outcomes of most simulation studies for laser annealing have been limited in that the simulations employed restrictive assumptions such as one laser shot, a fixed beam position or two-dimensional model. These conditions differ considerably from actual process conditions. The description above is already included along with major achievements in the last paragraph of the introduction. Please consider this point.

  • It is suggested to highlight the limitations of this study, suggested improvements of this work and future directions in the conclusion section. Also, the conclusion can be presented better than the present form with more findings.

Ans) The limitation of this study is the lack of validation of the numerical model. Additional concluding comments including the limitation and future direction of this study have been added to the conclusion section (line 326-332 in page 15).

Please note that the comments are intended merely to assist the authors in improving the manuscript and ensuring that published papers are of the highest quality. They are in NO WAY intended to discourage or demean the authors personally.

Reviewer 6 Report

This manuscript offers a concise yet compelling report on a numerical analysis of the temperature distribution during the laser annealing process of amorphous silicon (a-Si) used in a cell-over-periphery (COP) structure. A detailed theoretical modeling scheme is utilized to scrutinize temperature distribution. The chosen approaches, temporal profiles and regimes and corresponding considerations, and not least nice comparative context of results throughout the study, all contribute to the elaboration of a successful and reliable protocol for studying specific nanostructured materials. From practical point of view, the reported results bring new knowledge and certainly represent an original contribution in the present context.

The authors chose an adequate structure of the manuscript – an excellent point of departure for such a study. Finally, the authors provided a balanced realistic and nicely illustrated presentation of their results and corresponding analysis that is of much scientific and practical interest and adds new knowledge to the field.

In my opinion, the fine detailing in the present work, the insightful and balanced discussion of the results, as well as the very good figures, permit competent readers to utilize the manuscript as a guidance for future work. Consequently, this manuscript presents an efficient and beneficial basis for promoting and solving next step challenges in this field.

Moreover, the manuscript benefits from a clear motivation and it is an easy and informative read. The manuscript is also excellent in terms of clarity and accuracy of language.

The present manuscript is a significant contribution, this work once published would be quite useful as well as instructive and suggestive in terms of further studies and to a wider readership.

There are some minor issues with this already excellent manuscript that will need to be addressed before becoming suitable for publication, i.e., it can be considered for publication after a minor revision:

1: In the introduction, the authors partly miss the general scope and miss examples of appropriate theoretical work of structural and evolutionary properties of novel nanostructured systems, that are adequate yet compelling objects similar to their object of study of amorphous Si but using more basic and ab-initio approaches, like DFT. Examples include: The Journal of Physical Chemistry C 118 (2014) 5501-5509; Nanoscale 12 (37), 19470-19476. Such works can support by theoretical and ab initio methods the conceptualization as formatted in the present manuscript.

2: As reported in this manuscript expected temperatures at the interface between the a-Si and the SiO2 are found to be 1822 K, 1944 K, and 1958 K for overlap ratios of 0%, 25% and 50%, respectively. Any observations predictions about the bonding at the mentioned interface? This is conceptually important to many questions, to what extent bonding at the interface is thermally defined; maybe it will be appropriate to introduce in the text a more explicit discussion in this context.

3: The discussion in Sec. 3.4 about formation nucleation sites for grain growth is addressable by molecular dynamics simulations. Are the authors familiar with any such attempts to the material system in question here?

4: The authors may think for a shorter and more attractive title. Also, it is grammatically incorrect to begin every word in a title by a capital letter.

5: Spell-check and stylistic revision of the paper are still necessary. Some long sentences, misspellings, etc., are noticeable throughout the text.

Author Response

To editors and reviewers:

We really appreciate your valuable comments. We have finished the revision and also made responses to the comments of reviewers. The detail contents of the revision and response are described below (the edited parts are colored ‘red’ in the revised manuscript).  

Reviewer 6

Comments and Suggestions for Authors

This manuscript offers a concise yet compelling report on a numerical analysis of the temperature distribution during the laser annealing process of amorphous silicon (a-Si) used in a cell-over-periphery (COP) structure. A detailed theoretical modeling scheme is utilized to scrutinize temperature distribution. The chosen approaches, temporal profiles and regimes and corresponding considerations, and not least nice comparative context of results throughout the study, all contribute to the elaboration of a successful and reliable protocol for studying specific nanostructured materials. From practical point of view, the reported results bring new knowledge and certainly represent an original contribution in the present context.

The authors chose an adequate structure of the manuscript – an excellent point of departure for such a study. Finally, the authors provided a balanced realistic and nicely illustrated presentation of their results and corresponding analysis that is of much scientific and practical interest and adds new knowledge to the field.

In my opinion, the fine detailing in the present work, the insightful and balanced discussion of the results, as well as the very good figures, permit competent readers to utilize the manuscript as a guidance for future work. Consequently, this manuscript presents an efficient and beneficial basis for promoting and solving next step challenges in this field.

Moreover, the manuscript benefits from a clear motivation and it is an easy and informative read. The manuscript is also excellent in terms of clarity and accuracy of language.

The present manuscript is a significant contribution, this work once published would be quite useful as well as instructive and suggestive in terms of further studies and to a wider readership.

There are some minor issues with this already excellent manuscript that will need to be addressed before becoming suitable for publication, i.e., it can be considered for publication after a minor revision:

1: In the introduction, the authors partly miss the general scope and miss examples of appropriate theoretical work of structural and evolutionary properties of novel nanostructured systems, that are adequate yet compelling objects similar to their object of study of amorphous Si but using more basic and ab-initio approaches, like DFT. Examples include: The Journal of Physical Chemistry C 118 (2014) 5501-5509; Nanoscale 12 (37), 19470-19476. Such works can support by theoretical and ab initio methods the conceptualization as formatted in the present manuscript.

Ans) Thank you for your kind introduction to new literatures. The studies mentioned above handle quite interesting contents related to the nanostructured systems that could be adopted in various advanced semiconductor devices. However, theoretical works and ab initio methods introduced in the studies are unfamiliar to us (actually, we are not in the fields of material science, physic and chemistry). Therefore, we want to avoid mentioning the general scope of semiconductor technology (that we don’t know well) if possible. In the introduction, for the semiconductor structure and system, we only handled the technology evolution of a particular semiconductor device (NAND) because we wanted to describe more details about the application of the laser annealing for semiconductor field.

2: As reported in this manuscript expected temperatures at the interface between the a-Si and the SiO2 are found to be 1822 K, 1944 K, and 1958 K for overlap ratios of 0%, 25% and 50%, respectively. Any observations predictions about the bonding at the mentioned interface? This is conceptually important to many questions, to what extent bonding at the interface is thermally defined; maybe it will be appropriate to introduce in the text a more explicit discussion in this context.

Ans) As presented in our paper, numerical simulation to obtain temperature distribution was only conducted. Laser annealing experiment and the corresponding measurement and analysis of annealed samples (observation of the bonding interface between Si and oxide using SEM or TEM) were not performed. There may be some degradation (such as delamination) in the boding state of two layers because the laser annealing is high temperature process, and rapid heating and cooling are usually induced during the process. However, it is considered that without relevant data, describing the state of the bonding interface in this study could produce unclear discussion and some arguments. Please consider this point.

3: The discussion in Sec. 3.4 about formation nucleation sites for grain growth is addressable by molecular dynamics simulations. Are the authors familiar with any such attempts to the material system in question here?

Ans) Actually, we are not familiar with molecular dynamics simulations.

4: The authors may think for a shorter and more attractive title. Also, it is grammatically incorrect to begin every word in a title by a capital letter.

Ans) Title of the study has been changed to a shorter one, and as far as we know, using a capital letter for the first letter of every word in a title is the format of Materials.

5: Spell-check and stylistic revision of the paper are still necessary. Some long sentences, misspellings, etc., are noticeable throughout the text.

Ans) English editing for the whole text has been conducted.

Round 2

Reviewer 2 Report

1. Abstract: Describe more relevant results in the abstract. Mention the purpose for which this study was conducted.

2. The lengthy sentences may be split in to smaller sentence without change of its meaning.

3. Also, suggested to include the recent references in the introduction part.

4. The results and discussions part should be compared with the literature data. To redo the part of results and discussions by a systematic presentation of the results by which the readers of the articles manage to follow the article more easily..

5. Figure quality is poor throughout. To improve the quality of the figures. Enlarge the characters in the figures.

6. To correlate the results obtained with the results present in other works.

7. Conclusions should be short with important observations.

8. References are not written in unison. Some journals are abbreviated and some are not.

9. To pass the work in the materials page. Neither the format of the diary nor the size of the figures with the related legends are respected.

Author Response

To editors and reviewers:

We really appreciate your valuable comments. We have finished the 2nd revision and also made responses to the comments of a reviewer. The detail contents of the revision and response are described below (the edited parts are colored ‘red’ in the revised manuscript).  

Comments and Suggestions for Authors

  1. Abstract: Describe more relevant results in the abstract. Mention the purpose for which this study was conducted.

Ans) The abstract has been edited to include the purpose of the study and more relevant results.

  1. The lengthy sentences may be split in to smaller sentence without change of its meaning.

Ans) The lengthy sentences have been rewritten for a concise statement (see line 116-118, line 138-141, line 239-241, and line 278-280).

  1. Also, suggested to include the recent references in the introduction part.

Ans) Three recent references (ref. 15-17 in the revised manuscript) have been added to the introduction part. Due the addition of new references, the numbering for the references has been updated.

  1. The results and discussions part should be compared with the literature data. To redo the part of results and discussions by a systematic presentation of the results by which the readers of the articles manage to follow the article more easily.

Ans) We have included some comparative discussions for the results obtained in this study and the results of other works (see line 222-227 and line 258-262). Due the addition of new references, the numbering for the references has been updated.

  1. Figure quality is poor throughout. To improve the quality of the figures. Enlarge the characters in the figures.

Ans) The quality of all figures have been improved.

-. Increasing the resolution of figures and enlarging the characters in figuresà all figures

-. Adding a legend to a figure à Fig. 7

-. Relocation of figures à Figs. 3 and 6

-. Minor editing of figures (to improve visibility of figures) à all figures

  1. To correlate the results obtained with the results present in other works.

Ans) Some comparative discussions for the results obtained in this study and the results of other works were added to the results and discussion section, as commented above.

  1. Conclusions should be short with important observations.

Ans) The conclusion part has been shortened.

  1. References are not written in unison. Some journals are abbreviated and some are not.

Ans) All journal (also for conference or symposium) names have been abbreviated.

  1. To pass the work in the materials page. Neither the format of the diary nor the size of the figures with the related legends are respected.

Ans) We has improved the quality of figures, as commented above. For the manuscript format, we humbly request assistant editor to edit our paper to follow the format of Materials.

Round 3

Reviewer 2 Report

I consider that all the requested corrections and completions have been made successfully. I propose the paper for publication.